# Cation vacancy stabilization of single-atomic-site Pt$_1$/Ni(OH)$_x$ catalyst for diboration of alkynes and alkenes

Jian Zhang [1], Xi Wu[2], Weng-Chon Cheong [1], Wenxing Chen [1], Rui Lin[1], Jia Li [2], Lirong Zheng[3], Wensheng Yan[4], Lin Gu[5], Chen Chen[1], Qing Peng[1], Dingsheng Wang [1] & Yadong Li[1]

Development of single-atomic-site catalysts with high metal loading is highly desirable but proved to be very challenging. Although utilizing defects on supports to stabilize independent metal atoms has become a powerful method to fabricate single-atomic-site catalysts, little attention has been devoted to cation vacancy defects. Here we report a nickel hydroxide nanoboard with abundant Ni$^{2+}$ vacancy defects serving as the practical support to achieve a single-atomic-site Pt catalyst (Pt$_1$/Ni(OH)$_x$) containing Pt up to 2.3 wt% just by a simple wet impregnation method. The Ni$^{2+}$ vacancies are found to have strong stabilizing effect of single-atomic Pt species, which is determined by X-ray absorption spectrometry analyses and density functional theory calculations. This Pt$_1$/Ni(OH)$_x$ catalyst shows a high catalytic efficiency in diboration of a variety of alkynes and alkenes, yielding an overall turnover frequency value upon reaction completion for phenylacetylene of ~3000 h$^{-1}$, which is much higher than other reported heterogeneous catalysts.

[1] Department of Chemistry, Tsinghua University, 100084 Beijing, China. [2] Laboratory for Computational Materials Engineering, Division of Energy and Environment, Graduate School at Shenzhen, Tsinghua University, 518055 Shenzhen, China. [3] Beijing Synchrotron Radiation Facility, Institute of High Energy Physics, Chinese Academy of Sciences, 100049 Beijing, China. [4] National Synchrotron Radiation Laboratory, CAS Center for Excellence in Nanoscience, University of Science and Technology of China, 230029 Hefei, China. [5] Beijing National Laboratory for Condensed Matter Physics, Institute of Physics, Chinese Academy of Sciences, 100190 Beijing, China. These authors contributed equally: Jian Zhang, Xi Wu. Correspondence and requests for materials should be addressed to D.W. (email: wangdingsheng@mail.tsinghua.edu.cn) or to Y.L. (email: ydli@mail.tsinghua.edu.cn)

Single-atomic-site (SAS) heterogeneous catalysts have attracted much recent interest owing to their specific activity and maximum atom efficiency for low cost[1–10]. However, synthesis of such SAS catalysts is not trivial because isolated metal atoms are often very mobile and easy to sinter under realistic reaction conditions due to their high surface free energy[2,4]. For this reason, most available SAS catalysts must keep a low loading density of guest metals (usually <0.5 weight percent (wt%)) to resist their aggregation, and it remains a great challenge to improve the loading content in such catalysts for practical applications[5]. Exploiting defects on supports to enhance the interaction between individual metal atoms and the supports has been an effective strategy to fabricate SAS catalysts[11–22]. So far, much work has focused on oxygen vacancy defects on oxides and carbon vacancy defects on graphene[11–18]. Cation vacancies are another kind of classical defects but are comparatively little investigated in the research field of SAS catalysts[19–22], probably because of their difficult characterization and scarce suitable support materials with such defects[23]. Hydroxides are a large class of functional, environmentally friendly, and inexpensive host materials[24]. As far as we know, the cation vacancies on hydroxides have never been reported, and utilizing the defect-rich hydroxide to achieve a high metal-loading SAS catalyst has not been realized yet.

Boronic acids and their derivatives are versatile and useful compounds for various applications in organic synthesis[25], material science[26], and biomedicine[27]. Over the past decades, a broad variety of transition-metal-catalyzed protocols have been developed for the preparation of these compounds[28]. Among them, the diboration of carbon–carbon multiple bonds represents a straightforward and atom-economic strategy[29]. Since the first discovery of the Pt-catalyzed diboration of alkynes by Suzuki and Miyaura et al. in 1993, various homogeneous transition-metal catalysts have been successfully applied into the diboration of alkynes or alkenes[29–35]. However, up to now, the development of heterogeneous catalysts for such diboration reactions lags far behind the homogeneous catalysts with limited reported cases that include Pd/C[36], nanoporous-gold[37], Pt/TiO$_2$[38], and Pt/MgO[39]. To make matter worse, these heterogeneous catalysts are restricted in practical application for their low catalytic efficiency (overall turnover frequency (TOF$_{overall}$) upon reaction completion <50 h$^{-1}$). There is thereby an urgent need to prepare a new heterogeneous catalyst with better catalytic efficiency for diboration reactions. Given that the catalytically active components in these reported catalysts are all metal nanoparticles and downsizing metal particles to single atoms is ordinarily a great impetus to improve the performance of a catalyst[4,5], we expect that the rational design of SAS catalysts will offer exciting opportunities to achieve the ideal heterogeneous catalysts for diboration reactions.

Here we report that a defect-rich nickel hydroxide (Ni(OH)$_x$) nanoboards (NBs) supported SAS Pt catalyst (Pt$_1$/Ni(OH)$_x$) fabricated by a simple wet impregnation method. Notably, although there have been a few reports on the combination of nickel hydroxides with Pt nanoparticles, the construction of SAS Pt species on nickel hydroxides has never been achieved[40–42]. In this work, a new polycrystalline Ni(OH)$_x$ NBs are synthesized on a large scale via a one-pot solvothermal procedure. The abundant Ni$^{2+}$ vacancy defects on the Ni(OH)$_x$ NBs are shown to be critical for preparing Pt$_1$/Ni(OH)$_x$ with Pt loading up to 2.3 wt%. The as-prepared Pt$_1$/Ni(OH)$_x$ catalyst exhibits a good performance for the diboration of alkynes and alkenes. A TOF$_{overall}$ upon reaction completion much greater than that of all reported heterogeneous catalysts is demonstrated on Pt$_1$/Ni(OH)$_x$ in the diboration of alkynes.

## Results

**Synthesis and characterization of the Pt$_1$/Ni(OH)$_x$ catalyst.** To prepare the Pt$_1$/Ni(OH)$_x$ catalyst, a polycrystalline Ni(OH)$_x$ NB material was first synthesized on a large scale through a solvothermal reaction between nickel nitrate (Ni(NO$_3$)$_2$·6H$_2$O), urea, sodium bicarbonate (NaHCO$_3$), and tetrabutylammonium hydroxide (TBAH) in water/triethylene glycol mixed solvent (for details, see the Methods section). The typical transmission electron microscopic (TEM) image clearly illustrates that the as-synthesized samples display a uniformly NB morphology (Fig. 1a). Clear irregular crystal lattice fringes are observed on the NBs in the high-resolution TEM (HR-TEM) image (Fig. 1b), indicating the polycrystalline structure of the sample, which is further proved by the selected-area electron diffraction pattern and X-ray diffraction (XRD) pattern (Supplementary Figs. 1 and 2). X-ray photoelectron spectroscopy (XPS) and Fourier transform infrared spectroscopy (FT-IR) of this polycrystalline NBs both exhibit features that are typical nickel hydroxide (Supplementary Fig. 3). To the best of our knowledge, this one-dimensional polycrystalline nanostructures of nickel hydroxide are newly synthesized by our work, which are more challenging in synthesis compared with common nickel hydroxide nanosheets[43,44].

The Pt$_1$/Ni(OH)$_x$ catalyst was then prepared with the as-synthesized Ni(OH)$_x$ NB material as a support by a wet impregnation method, which stands for an easy-handling, straightforward, and low-cost pathway to synthesize catalysts[45,46]. Hexachloroplatinic acid (H$_2$PtCl$_6$) was introduced into an ethanol dispersion of Ni(OH)$_x$ NBs to allow the adsorption of Pt precursors. The mixture was then centrifuged and the recovered solid was reduced with hydrogen to provide the Pt$_1$/Ni(OH)$_x$ catalyst (for details, see the Methods section). Scanning transmission electron microscopy (STEM) images and XRD detections of the obtained Pt$_1$/Ni(OH)$_x$ reveal that no formation of Pt nanoparticles are observed on Ni(OH)$_x$ NBs, even with the

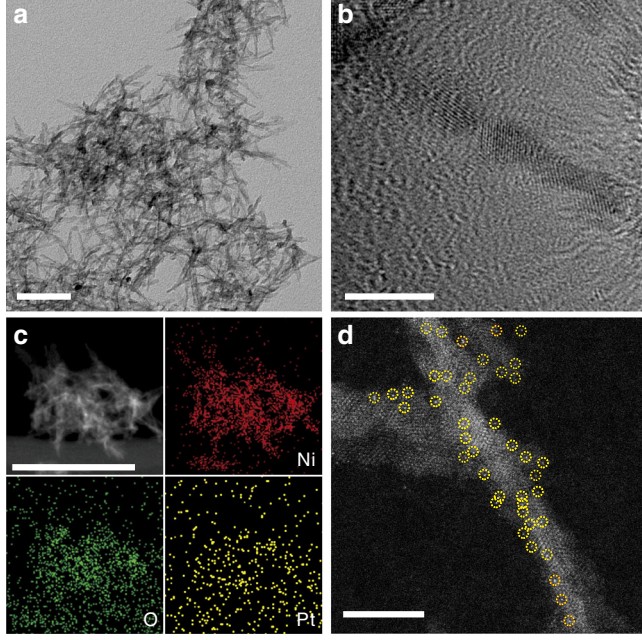

**Fig. 1** Characterization of Ni(OH)$_x$ NBs and the Pt$_1$/Ni(OH)$_x$ catalyst. **a** TEM image of Ni(OH)$_x$ NBs. Scale bar, 50 nm. **b** HR-TEM image of a Ni(OH)$_x$ nanoboard. Scale bar, 20 nm. **c** EDX elemental mapping analysis of the Pt$_1$/Ni(OH)$_x$ catalyst. Scale bar, 100 nm. **d** Representative AC HAADF-STEM image of the Pt$_1$/Ni(OH)$_x$ catalyst. The yellow circles were drawn around SAS Pt. Scale bar, 10 nm

loading amount of Pt as high as 2.3 wt% as analyzed by inductively coupled plasma optical emission spectrometry (ICP-OES) (Supplementary Fig. 4). Further energy-dispersive X-ray (EDX) elemental mapping analysis confirms that Pt species are evenly dispersed in Pt$_1$/Ni(OH)$_x$ (Fig. 1c). To verify the SAS Pt species on the Ni(OH)$_x$ NBs, we performed the aberration-corrected high-angle annular dark-field STEM (AC HAADF-STEM) measurements on Pt$_1$/Ni(OH)$_x$ (Fig. 1d). It is clear that all the Pt species exist exclusively at isolated single atomic sites; neither subnanometer clusters nor nanoparticles are detected.

The presence of SAS Pt can be further confirmed by X-ray absorption spectrometric (XAS) studies. Figure 2a represents the extended X-ray absorption fine structure (EXAFS) spectrum of Pt$_1$/Ni(OH)$_x$ and the reference spectra of Pt foil and PtO$_2$ at the Pt L$_3$-edge using a Fourier transform (for corresponding EXAFS in K-space, see Supplementary Fig. 5). There is one prominent peak at ~1.6 Å from the Pt–O contribution and a relatively weak peak at ~2.9 Å from the Pt–Ni contribution but no peak at ~2.6 Å from the Pt–Pt contribution, confirming the sole presence of SAS Pt in the Pt$_1$/Ni(OH)$_x$ catalyst. Moreover, the oxidation state of these SAS Pt is determined by the X-ray absorption near-edge structure (XANES) spectra, as shown in Fig. 2b. The white-line intensities in the spectra reflect the oxidation state of Pt in different samples, so the white-line intensity of Pt$_1$/Ni(OH)$_x$, which is close to that of PtO$_2$, implies that the SAS Pt in the Pt$_1$/Ni(OH)$_x$ catalyst still remain in a high oxidation state even after the reduction by hydrogen.

**Catalytic performance evaluation for diboration reactions**. We next investigated the catalytic activity of the as-prepared Pt$_1$/Ni(OH)$_x$ for diboration reactions. Initially, the diboration of phenylacetylene (**1a**) with bis(pinacolato)diboron (B$_2$pin$_2$) (**2a**) was chosen as a model reaction (Supplementary Table 1, entry 1). To our delight, the Pt$_1$/Ni(OH)$_x$ displayed a high activity and selectivity for this reaction. The conversion of phenylacetylene attained 97% within 20 min at a molar ration of 1:10$^3$ (Pt: phenylacetylene), and bare of other by-products like hydroborylated product were observed by gas chromatography–mass spectrometer (GC-MS) analysis (Supplementary Fig. 6). The calculated TOF$_{overall}$ value of Pt$_1$/Ni(OH)$_x$ upon this reaction completion can reach a high level as ~3000 h$^{-1}$, much higher than that of other heterogeneous catalysts reported previously. In contrast, no reaction occurred when using Ni(OH)$_x$ NBs as the catalyst without Pt (Supplementary Table 1, entry 2). The catalyst can be reused at least five times without any loss of selectivity although the activity has a slight decay (Supplementary Fig. 7). STEM and AC HAADF-STEM characterizations reveal that no morphology changes and Pt components are still dispersed at isolated single

atomic sites in the recovered catalyst (Supplementary Fig. 8). Moreover, the expansion of the reaction scale has no effect on the catalytic efficiency of this Pt$_1$/Ni(OH)$_x$ catalyst for such diboration reactions (Supplementary Fig. 9). We further investigated the substrate scope of the diboration reactions to study the influence of substrate categories on the catalytic efficiency of Pt$_1$/Ni(OH)$_x$. As shown in Fig. 3, the aryl alkynes bearing electron-donating groups (R = Me, OMe) can react smoothly with B$_2$pin$_2$ over Pt$_1$/Ni(OH)$_x$ at the same conversion rate of phenylacetylene, affording the corresponding products **3ba** and **3ca** in excellent yields. When the substituents are electron-withdrawing groups (R = Cl, Br, NO$_2$) on the aryl ring, however, a longer reaction time is required to give the target molecular **3da–3fa** in high yields. Differently from substituent types, substituent positions have no influence on the catalytic efficiency of Pt$_1$/Ni(OH)$_x$, whether *meta*-methyl or *ortho*-methyl substituted phenylacetylene can completely transform into the products **3ga** and **3ha** without any loss in the reaction rate. As for diboration of internal aryl alkynes like diphenylacetylene, Pt$_1$/Ni(OH)$_x$ suffers from a relatively low catalytic activity, although a complete conversion of the substrate and a quantitative selectivity of diborylated product **3ia** can be also achieved. To our delight, different kinds of aliphatic alkynes are appropriate for the diboration reactions over Pt$_1$/Ni(OH)$_x$ as well, furnishing the desired products **3ja–3la** with the similar reaction efficiency to the terminal aryl alkynes. Besides the various alkynes, different boronate esters like bis(neopentylglycolate)diboron (B$_2$neop$_2$) can also work well with phenylacetylene to provide the product **3ab** in an excellent yield and selectivity at the same reaction rate of B$_2$pin$_2$. Even more, alkenes were also chosen as substrates to evaluate the catalytic performance of Pt$_1$/Ni(OH)$_x$ for diboration reactions. It shows that the diboration of styrene and 1-octene catalyzed by Pt$_1$/Ni(OH)$_x$ can proceed well and provide a selectivity to products **3ma** and **3na** of 99% at the conversion level of 90% and 86%, respectively, although the reaction rates are lower than that of alkynes with similar molecular structures.

## Discussion

Apparently, the good catalytic performance of Pt$_1$/Ni(OH)$_x$ derives from the high loading content of Pt at isolated single atomic sites. It is noteworthy that such a high loading density of SAS catalyst is quite difficult in fabrication through the wet impregnation method[47–50]. The impregnated metal precursors generally adsorb on the surface of supports and thus tend to aggregate to form clusters or nanoparticles easily during the post-treatment processes[1]. For comparison, a conventional perfect Ni(OH)$_2$ material was synthesized and impregnated with H$_2$PtCl$_6$ with a lower Pt loading at 0.9 wt% (as determined by ICP-OES)

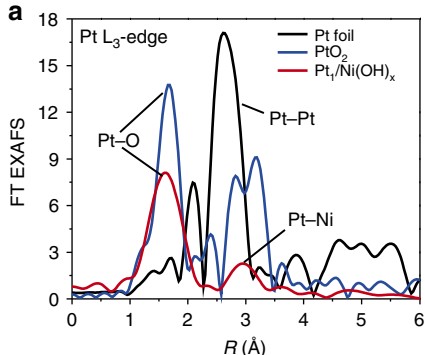
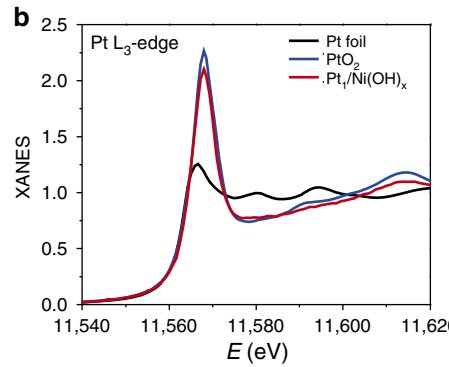

**Fig. 2** X-ray absorption spectrometric studies of the Pt$_1$/Ni(OH)$_x$ catalyst. **a** Fourier transform EXAFS spectrum of the Pt$_1$/Ni(OH)$_x$ catalyst in comparison with PtO$_2$ and Pt foil at the Pt L$_3$-edge. **b** XANES spectra at the Pt L$_3$-edge of the Pt$_1$/Ni(OH)$_x$ catalyst, PtO$_2$, and Pt foil

**Fig. 3** Substrate scope of diboration reactions over the Pt$_1$/Ni(OH)$_x$ catalyst. Standard reaction conditions: substrate **1** (0.50 mmol) and **2** (0.50 mmol), Pt$_1$/Ni(OH)$_x$ catalyst, Pt/substrate = 0.1%, mesitylene (2.0 mL) as solvent, $T$ = 120 °C, $t$ = 0.3 h. Conversion are determined by gas chromatography (GC) analysis with dodecane as internal standard. Selectivities are determined by GC-MS analysis. $^a$ $t$ = 1.0 h. $^b$ $t$ = 6.0 h and substrate **2** (0.75 mmol) was used

under the same conditions as that of Ni(OH)$_x$ NBs (Supplementary Fig. 10). As expected, in the obtained Pt/Ni(OH)$_2$ sample, numbers of Pt nanoparticles are observed clearly on the perfect Ni(OH)$_2$ by HR-TEM (Supplementary Fig. 11), which results in a relatively low catalytic efficiency of Pt/Ni(OH)$_2$ for diboration reactions (Supplementary Table 1, entry 3 and 4). This visible difference indicates that Ni(OH)$_x$ NBs have a stronger interaction with isolated Pt atoms than the perfect Ni(OH)$_2$ to prevent the formation of Pt clusters or nanoparticles.

To explore the nature of this strong interaction, we first carried out the EXAFS spectrometry analysis to probe the atomic structure of these two different nickel hydroxides. As shown in Fig. 4a, the Ni K-edge Fourier-transformed EXAFS spectrum of Ni(OH)$_x$ NBs exhibit an apparent difference in spectral shape compared

with that of the perfect Ni(OH)$_2$, implying the different local atomic arrangement and a defective structure of Ni(OH)$_x$ NBs[51]. Further EXAFS fitting analysis revealed that the values of Debye–Waller factor ($\sigma^2$) for the first Ni–O and Ni–Ni shells of Ni(OH)$_x$ NBs are both higher than that of the perfect Ni(OH)$_2$, suggesting a higher degree of disorder in Ni(OH)$_x$ NBs, which is in accord with the polycrystalline structure of Ni(OH)$_x$ NBs (Supplementary Table 2). More importantly, the coordination number ($N$) of the first Ni–Ni shell of Ni(OH)$_x$ NBs is about 4.8, which is lower than that of the perfect Ni(OH)$_2$ (~6.2), whereas their coordination numbers of the first Ni-O shell are nearly same (~6.0), indicating the formation of Ni$^{2+}$ vacancies in Ni(OH)$_x$ NBs. Many studies show that the formation of Ni$^{2+}$ vacancies will lead some Ni$^{2+}$ ions to transform into Ni$^{3+}$ ions due to the

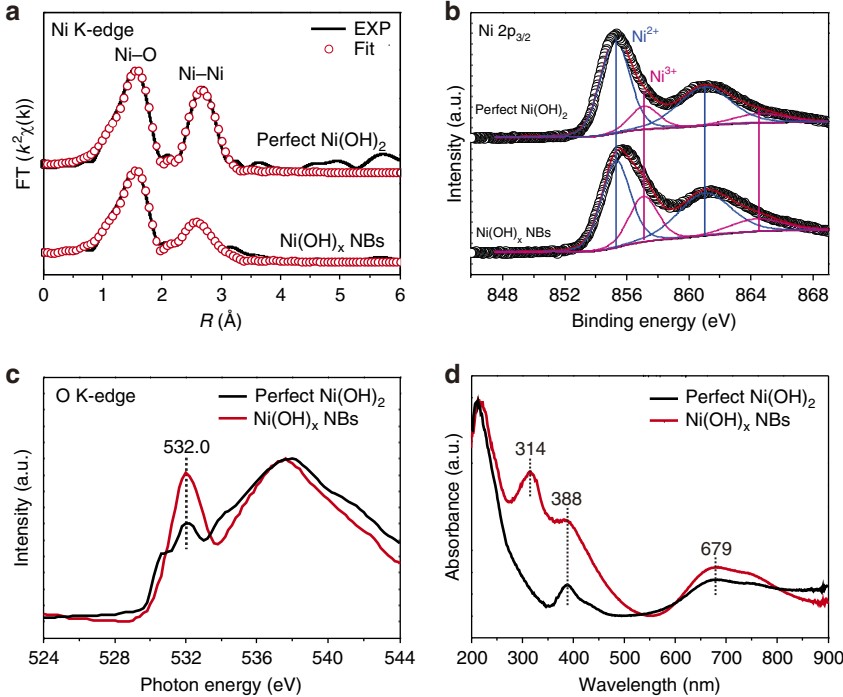

**Fig. 4** Investigation of cation vacancies on Ni(OH)$_x$ NBs. **a** Ni K-edge Fourier transform EXAFS spectra and corresponding fitting analysis for Ni(OH)$_x$ NBs and the perfect Ni(OH)$_2$. **b** XPS analysis of Ni(OH)$_x$ NBs and the perfect Ni(OH)$_2$ in the Ni 2p$_{3/2}$ region. **c** O K-edge sXAS spectra of Ni(OH)$_x$ NBs and the perfect Ni(OH)$_2$. **d** UV-Vis DRS spectrum of Ni(OH)$_x$ NBs compared with the perfect Ni(OH)$_2$

charge neutrality[52]. Hence, we carried out XPS measurements to detect the Ni$^{3+}$ ions in the Ni(OH)$_x$ NBs. Figure 4b displays the representative XPS spectrum in Ni 2p$_{3/2}$ region of Ni(OH)$_x$ NBs and the perfect Ni(OH)$_2$, which can be deconvoluted into four peaks. The signal of Ni$^{3+}$ ions can be clearly distinguished from that of Ni$^{2+}$ ions (centered at 855.3 eV and 861.0 eV) with higher binding energies at 857.2 eV and 864.7 eV, respectively, which correspond with the data reported[53–55]. Distinctly, unlike the perfect Ni(OH)$_2$, Ni(OH)$_x$ NBs display a stronger signal of Ni$^{3+}$ ions, manifesting the possession of more Ni$^{3+}$ ions in the Ni (OH)$_x$ NBs. The proof of Ni$^{3+}$ ions in Ni(OH)$_x$ NBs is garnered from the soft XAS (sXAS) analysis, too. As shown in Fig. 4c, the obviously increased intensity at 532.0 eV (Ni–O interaction) at O K-edge of Ni(OH)$_x$ NBs relative to that of the perfect Ni(OH)$_2$ suggests that electrons transfer intensively from oxygen to nickel, which is consistent with the presence of Ni$^{3+}$ ions[56–58]. In addition, the ultraviolet-visible diffuse reflectance spectroscopy (UV-Vis DRS) experiments also evidence the Ni$^{3+}$ ions in Ni (OH)$_x$ NBs. As can be seen from Fig. 4d, besides the two absorption bands of Ni$^{2+}$ ions at 388 nm and 679 nm in the both two samples, a unique absorption band of Ni(OH)$_x$ NBs appears at 314 nm and is characteristic of Ni$^{3+}$ ions[59,60]. On the basis of all above evidences, we conclude that abundant Ni$^{2+}$ vacancies exist on the Ni(OH)$_x$ NBs and induce the strong interaction with isolated Pt atoms.

To gain more insight into the interaction between Ni$^{2+}$ vacancies and isolated Pt atoms, density functional theory (DFT) calculations were conducted to verify the different formation energies of the isolated Pt atoms loaded on the Ni(OH)$_2$ with and without Ni$^{2+}$ vacancies (for details, see Supplementary Methods section). As can be seen from Fig. 5, the Pt atom adsorbed on the Ni(OH)$_2$ with Ni$^{2+}$ vacancies displays a formation energy at −3.89 eV, which is much lower than that of the Pt atom adsorbed on the Ni(OH)$_2$ without Ni$^{2+}$ vacancies (at −0.72 eV). For the Ni (OH)$_2$ with Ni$^{2+}$ vacancies, the most stable adsorption site for the Pt atom is found to be the Ni$^{2+}$ vacancy site as well as the three-

fold hollow site of the oxygen atoms, and the Pt atom is fixed by the three top oxygen atoms near to the Ni$^{2+}$ vacancy according to the charge density difference (Fig. 5a). In contrast, the Pt atom on the Ni(OH)$_2$ without Ni$^{2+}$ vacancies tends to locate at the site slightly deviated from three-fold hollow site of oxygen atoms, which is caused by the competition between the strong interaction between the Pt atom and three top oxygen atoms and the electrostatic repulsion between positive charged Pt and Ni atoms (Fig. 5b). Furthermore, the oxidation states of isolated Pt atoms anchored on the Ni(OH)$_2$ with and without Ni$^{2+}$ vacancies were also estimated by evaluating Bader charges of the Pt atoms in the film and by normalizing them to Bader charges of PtO$_2$ (for details, see Supplementary Methods section). As a result, the oxidation state of the Pt atom on the Ni(OH)$_2$ with Ni$^{2+}$ vacancies is +3.55, which is higher than that on the Ni(OH)$_2$ without Ni$^{2+}$ vacancies (+2.70) and very compatible with the aforementioned XANES data of Pt$_1$/Ni(OH)$_x$ in Fig. 2b. This higher oxidation state illustrates the increase of charge transfer from the support to the Pt atoms[4]. In terms of these DFT calculation results and the XANES data, it is convinced that the Ni$^{2+}$ vacancies play a vital role in the stabilization of isolated Pt atoms deposited on the Ni(OH)$_x$ by eliminating the spatial segregation between the Pt atoms and uncoordinated O atoms, as well as decreasing the formation energy of the Pt atoms through promoting charge transfer from Ni(OH)$_x$ to them. Further DFT calculations on the catalytic mechanism of the Pt$_1$/Ni(OH)$_x$ catalyst for diboration reactions even disclosed that the Ni$^{2+}$ vacancies not only play an important role in locating isolated Pt atoms but also are conducive to the diboration reactions because the low-coordination oxygen atoms at the vacancy site around the located Pt atoms benefit the dissociation of B–B bonds (for details, see Supplementary Methods section).

In summary, we report that a defect-rich Ni(OH)$_x$ NBs supported SAS Pt catalyst with remarkable performance in diboration reactions. The Ni(OH)$_x$ NBs with a polycrystalline structure are newly synthesized and successfully loaded with SAS Pt to a high

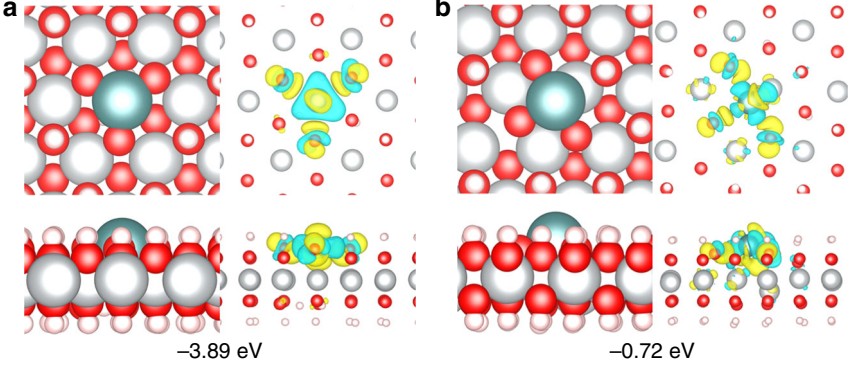

**Fig. 5** Studies of the interaction between Ni$^{2+}$ vacancies and isolated Pt atoms. Top and side views of the most stable structure and charge density difference for the Pt atom adsorbed on the Ni(OH)$_2$ with Ni$^{2+}$ vacancies (**a**) and without Ni$^{2+}$ vacancies (**b**). The cyan, gray, red, and white balls refer to Pt, Ni, O, and H atoms, respectively. For charge density difference, yellow (blue) corresponds to charge accumulation (depletion) plotted with an isovalue of ±0.01 e Å$^{-3}$

content as 2.3 wt% just through a simple wet impregnation method. Different from common perfect Ni(OH)$_2$, this defective Ni(OH)$_x$ is abundant in Ni$^{2+}$ vacancy defects that play a key role in stabilizing the Pt atoms at single atomic sites via an enhanced charge-transfer mechanism. The as-fabricated Pt$_1$/Ni(OH)$_x$ catalyst displays a good activity and selectivity in diboration of various alkynes and alkenes, and the greatest TOF$_{overall}$ value upon reaction completion can reach up to around 3000 h$^{-1}$, which is much higher than other heterogeneous catalysts reported in the literatures. Finally, our work suggests that SAS catalysts might open up new opportunities in heterogenization of homogeneous catalytic reactions for improved activity, and making use of the cation vacancy defects on supports to anchor the guest metal atoms would become a valid approach to prepare these SAS catalysts.

## Methods

**Synthesis of Ni(OH)$_x$ NBs.** Ni(NO$_3$)$_2$·6H$_2$O (2.5 mmol, 725.0 mg) was dissolved in deionized water (10 mL) and triethylene glycol (20 mL) and then mixed with an aqueous solution of urea (5.0 mmol, 300.0 mg), TBAH (25% aqueous solution, 0.8 mmol, 0.8 mL), and NaHCO$_3$ (1.5 mmol, 126.0 mg) in deionized water (10 mL). After vigorous stirring for 15 min at ambient temperature, the mixture was transferred into a 50-mL Teflon-lined stainless-steel autoclave and heated at 120 °C for 12 h. The green product was collected via centrifugation and further washed with deionized water and ethanol for two times, respectively. After drying in vacuum oven, the Ni(OH)$_x$ NBs were used for characterization and further preparation.

**Synthesis of the Pt$_1$/Ni(OH)$_x$ catalyst.** The as-synthesized Ni(OH)$_x$ NBs (100.0 mg) were first dispersed in 20 mL ethanol under ultrasonic vibration. A H$_2$PtCl$_6$ solution (6.3 mg in 5 mL ethanol) was next added dropwise into the Ni(OH)$_x$ NBs dispersion under stirring at ambient temperature. After continuous stirring overnight, the suspension was centrifuged. The recovered solid was then dried in vacuum oven and reduced in 5% H$_2$/N$_2$ at 100 °C for 2 h to afford the Pt$_1$/Ni(OH)$_x$ catalyst for further characterization and catalysis test.

**Synthesis of the perfect Ni(OH)$_2$ and Pt/Ni(OH)$_2$.** The conventional perfect Ni(OH)$_2$ were prepared by following a modified synthetic method in the literature[61]. In a typical procedure, Ni(NO$_3$)$_2$·6H$_2$O (5.0 mmol, 1.45 g) and urea (20.0 mmol, 1.20 g) were dispersed in a mixture containing deionized water (10 mL) and triethylene glycol (70 mL) under vigorous stiring. The final solution was sealed in a 100-mL Teflon-lined stainless-steel autoclave and heated at 100 °C for 6 h. The perfect Ni(OH)$_2$ were obtained by centrifugation and further washed with deionized water and ethanol for two times, respectively. Finally, they were dried at 70 °C for 12 h before characterization and further preparation. With the as-prepared perfect Ni(OH)$_2$ as the support, the Pt/Ni(OH)$_2$ was synthesized through the same procedure of the fabrication of Pt$_1$/Ni(OH)$_x$ catalyst aforementioned but with a more dilute H$_2$PtCl$_6$ solution (2.1 mg in 5 mL ethanol).

**Measurements of diboration reactions.** All manipulations were carried out using standard Schlenk techniques. Unless otherwise noted, analytical grade solvents and commercially available reagents were used as received. In a typical

procedure, alkynes or alkenes (0.5 mmol), B$_2$pin$_2$ (0.5 mmol), and Pt$_1$/Ni(OH)$_x$ (Pt/ substrate = 0.1%) were placed in a Shlenck tube equipped with a stir bar, and then mesitylene (2.0 mL) was injected and the mixture was stirred at 120 °C for the corresponding reaction time. After the reaction was completed, the reaction mixture was analyzed by GC and GC-MS with dodecane as the internal standard. The overall TOF value was measured upon completion of reactions and the calculation of it was based on the total Pt loading in the catalyst.

**Characterization.** TEM images were taken from a Hitachi H-800 transmission electron microscope operated at 100 kV. HR-TEM, STEM, and EDX elemental mapping characterizations were carried out on a JEOL JEM-2100F field emission transmission electron microscope operated at 200 kV. The AC-HAADF STEM characterization was conducted on a Titan 80–300 scanning/transmission electron microscope operated at 300 kV, equipped with a probe spherical aberration corrector. XPS data were collected from a Thermo Fisher Scientific ESCALAB 250Xi XPS System, and the binding energy of the C1s peak at 284.8 eV was taken as an internal reference. The O K-edge sXAS spectra were collected at BL12B station of National Synchrotron Radiation Laboratory (NRSL) in Hefei, China. EXAFS spectra at Pt L$_3$-edge and Ni K-edge and the XANES spectra at Pt L$_3$-edge were all collected at the 1W1B station in Beijing Synchrotron Radiation Facility in transmission mode using a fixed-exit Si (111) double crystal monochromator. The incident X-ray beam was monitored by an ionization chamber filled with N$_2$, and the acquired EXAFS data were processed according to the standard procedures using the ATHENA module implemented in the IFEFFIT software packages. XRD data were acquired from a Rigaku RU-200b X-ray powder diffractometer with Cu Kα radiation (λ = 1.5406 Å). ICP-OES measurements were conducted on a Thermo Fisher iCAP™ 7000 Series ICP-OES analyzer. FT-IR spectroscopy was performed on a Bruker V70 infrared spectrometer in the frequency of 600–4000 cm$^{-1}$. UV-Vis DRS spectra were acquired from a Hitachi U-3900 UV–vis spectrophotometer. The GC analysis was conducted on a Thermo Trace 1300 series GC with a FID detector using a capillary column (TR-5MS, from Thermo Scientific, length 30 m, i.d. 0.25 mm, film 0.25 μm). The GC-MS analysis was carried out on a ISQ GC-MS with a ECD detector (Thermo Trace GC Ultra) using a capillary column (TR-5MS, from Thermo Scientific, length 30 m, i.d. 0.25 mm, film 0.25 μm). $^1$H nuclear magnetic resonance (NMR) and $^{13}$C NMR data were recorded with a Bruker Advance III (400 MHz) spectrometer. High-resolution exact mass measurements were performed on Thermo Scientific Q Exactive mass spectrometer. The detailed characterization data of products in the article are present in the Supplementary Methods section, and for the corresponding NMR spectra, see Supplementary Figs. 14–43.

**Data availability.** The data supporting this study are available from the authors upon reasonable request.

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

## Acknowledgements

This work was supported by the China Ministry of Science and Technology under Contract of (2016YFA (0202801), 2014CB932400, 2017YFB0701600) and the National Natural Science Foundation of China (21521091, 21390393, U1463202, 21471089, 21671117, 51232005) and Shenzhen Projects for Basic Research (Grant Nos. KQCX20140521161756227, JCYJ20170412171430026).

## Author contributions

J.Z. performed the experiments, collected and analyzed the data, and wrote the paper. X.W. and J.L. conducted the density functional theory calculation and analysis. W.-C.C. and R.L. assisted in HR-TEM, STEM, and EDX elemental mapping characterizations. W. C. and L.Z. helped with XANES and EXAFS spectrometry analyses. W.Y. helped with the sXAS analysis. L.G. assisted in the AC HAADF-STEM characterization. C.C. and Q.P. helped with data analyses and discussions. D.W. and Y.L. conceived the experiments, planned synthesis, analyzed results, and wrote the paper.

## Additional information

**Competing interests:** The authors declare no competing interests.

