## [Peer Review File · Nature Communications]

Editorial Note: Parts of this peer review file have been redacted as indicated to remove third-party material where no permission to publish could be obtained.

PEER REVIEW FILE

Reviewers' comments:

Reviewer #1 (Remarks to the Author):

This manuscript describes the synthesis, detailed characterization and theoretical investigation of Ni(OH)_x nanobelt (as the authors call it) and its use to atomically disperse Pt over it as a catalyst support. To my knowledge the concept of the specific cationic vacancy assisted Pt dispersion is new and remarkable catalytic results are reported with reasonable characterization of the materials. In principle, the novelties are high and it can be of high impact to the community. Having written that, I see that there are several points to be improved before the manuscript can be positively evaluated in the journal under consideration. More detailed comments are listed below.

- What is the motivation to call the structure “nanobelt”? I believe there are several other ways to call the structure (also from the literature on nickel hydroxides).
- There are numerous reports on nanostructured Ni(OH)₂, including Pt/Ni(OH)₂ (which should be well described in the manuscript), especially as electrocatalysts. Have the authors tried to synthesize and test the reported nanostructured Ni(OH)₂ for comparison? Is the reported nanostructured material better than those reported in literature?
- Generally speaking the structure determination (or the discussion given) of Ni(OH)_x with unique cationic defects is not fully convincing. For example what can one say from the SAED pattern Figure S1? I do not know how the authors conclude the poly-“crystalline” structure from the results. Actually, XANES and EXAFS at Ni should be used to clarify this point further, in my opinion. Also the XRD patterns (Fig. 3b) should be well discussed in comparison to that of perfect-Ni(OH)_x case (Fig. 7b) where one could extract the information about preferential orientation of the NB. Also, Raman could add further information about the materials.
- (EXAFS) The FT of the EXAFS is shown, but the quality of the analysis is not clear from the data shown. The spectra in k-space should be presented in ESI to evaluate the quality.
- Catalytic test using only the Ni(OH)_x NB support should be evaluated. Also, catalytic results using Pt supported on “perfect” Ni(OH)_x under the same condition as well as only support are missing.
- IR spectrum of “perfect” Ni(OH)_x?
- What was the motivation to use the specific Pt precursor?
- XP spectrum of Ni(OH)_x NB is shown in Fig. 3b and that of “perfect”-Ni(OH)_x in Fig. S9. The deconvoluted peaks look quite different, whereas the spectra look quite similar. It is speculated that different deconvolution procedures may have been used, causing the difference. It would be necessary to show the two spectra by overlaying them to be precise about the interpretation.

- (Fig. 3c & page 11. Line 188) XPS signal should neither be “absorption” nor “adsorption”.
- (Fig. 3c) The enhanced intensity of 532 eV is noted, but Fig. S2 is discussing about 531.0 eV assigned to oxygen atoms with slightly different origin (or expressed in different nuances). Clarification is needed.
- (page 3, line 54) “crabon” should be “carbon” I guess.

Reviewer #2 (Remarks to the Author):

The authors prepared cation-rich Ni(OH)_x nanobelts and used them to support single-atom Pt species for catalytic diboration of alkynes and alkenes. They found very high catalytic activity for the reaction and attributed it to the strong interaction of Pt at the cation vacancies of the support. They further characterized their catalysts with TEM, XPS, EXAFS, UV-vis, etc. They also performed DFT calculations to show the strong interaction between Pt and the cation-rich Ni(OH)_x nanobelts. I have the following comments for the authors to address before I can recommend its publication.

1. The authors' claim that their catalyst's turnover frequency is “60-fold higher than other reported heterogeneous catalysts” is very misleading because the reactions tested in “other reported heterogeneous catalysts” are very different from the present diboration reaction. For an apple-to-apple comparison, the authors should test the same reaction, e.g. CO oxidation, CO₂ hydrogenation, or other hydrogenation reactions tested before.
2. Can the authors show the results for the control, i.e., the cation-rich Ni(OH)_x nanobelts as a catalyst without Pt?
3. The DFT results of stronger binding of Pt on the cation-rich Ni(OH)_x nanobelts are expected, so they add limited value to the present work. It'd be more valuable if the authors could use DFT to further reveal the catalytic mechanism.

Reviewer #3 (Remarks to the Author):

This article describes a synthesis of diborylated alkenes or alkanes using heterogeneous catalysis.

It is interesting. It needs some proof reading to avoid errors such as "crabon carbon" and "dibration."

For publication, apart from the English revision, I would expect to see the following:

I) Table 1: Proper characterisation of the diborylated products. Yields quoted are by GC and as conversions. I would like to see both NMR characterisation (¹H, ¹³C, with scanned spectra in the ESI) and isolated (after chromatography) yields on > 1 mmol scale reactions.

I'd say accept with minor corrections.

II) There are only 5 examples in Table 1, which is hardly demonstrating reaction scope, substrate, functional group tolerance. I'd expect to see >10 examples including other boronate esters, other alkynes including more alkyl, heterocyclic, sterically and electronically diverse aryl derivatives to show yields vs steric and electronic effects.

III) Many homogeneous reference citations are missing-it would be nice to cite some recent reviews e.g. *Coord. Chem. Rev.* 2017, 336, 54; *Chem. Rev.* 2016, 116, 9091–9161.

Responses to Reviewers:

The reviewers' comments are laid out below in *Italic font* and specific concerns have been numbered. Our responses are given in normal font and changes/additions to the manuscript and supplementary information are highlighted by using **blue colored text**.

Following is our point to point answer to all the comments:

Reviewer #1:

This manuscript describes the synthesis, detailed characterization and theoretical investigation of Ni(OH)_x nanobelt (as the authors call it) and its use to atomically disperse Pt over it as a catalyst support. To my knowledge the concept of the specific cationic vacancy assisted Pt dispersion is new and remarkable catalytic results are reported with reasonable characterization of the materials. In principle, the novelties are high and it can be of high impact to the community. Having written that, I see that there are several points to be improved before the manuscript can be positively evaluated in the journal under consideration. More detailed comments are listed below.

Comment 1. *What is the motivation to call the structure “nanobelt”? I believe there are several other ways to call the structure (also from the literature on nickel hydroxides).*

Response: We sincerely appreciate your valuable suggestion. After referring to many literatures

on nickel hydroxides and other nanocrystal materials, we found out that the boardlike nanostructure of Ni(OH)₂ (*J. Phys. Chem. B* **109**, 7654-7658 (2005)) has the most similar morphology to our synthesized Ni(OH)_x material (Fig. R1). Therefore, we have replaced the “nanobelt” with “nanoboard” to call the structure of Ni(OH)_x in this work.

Comment 2. There are numerous reports on nanostructured Ni(OH)₂, including Pt/Ni(OH)₂ (which should be well described in the manuscript), especially as electrocatalysts. Have the authors tried to synthesize and test the reported nanostructured Ni(OH)₂ for comparison? Is the reported nanostructured material better than those reported in literature?

Response: Thank you very much for this comment. We have added the following sentence to describe the reported Pt/Ni(OH)₂:

p. 4 “Notably, although there have been a few reports on the combination of nickel hydroxides with Pt nanoparticles, the construction of ISAS Pt species on nickel hydroxides has never been achieved³⁷⁻³⁹”

In addition, we synthesized two different reported nanostructured Ni(OH)₂, one of which is a typical α -type Ni(OH)₂ (denoted as Ni(OH)₂(A), *Sci. Rep.* **4**, 5787 (2014).) and the other is a typical β -type Ni(OH)₂ (denoted as Ni(OH)₂(B), *Chem. Phys. Lett.* **405**, 159-164 (2005).) (Fig. R2(a-d)). For comparison, these two Ni(OH)₂ were used as supports to load Pt metal under the same condition of our Ni(OH)_x NBs (for details, see the Methods section in the manuscript), and the as-synthesized catalysts were denoted as Pt/Ni(OH)₂(A) and Pt/Ni(OH)₂(B). As shown in Fig. R2(e and f), **apparent Pt nanoparticles** can be observed on the both Ni(OH)₂(A) and Ni(OH)₂(B) in the STEM images. This phenomenon further indicates the advantage of our synthesized Ni(OH)_x NBs material to fabricate the supported isolated single-atomic-site Pt catalyst.

Figure R2. (a) TEM image of the as-prepared Ni(OH)₂(A) sample. Scale bar, 200 nm. (b) TEM image of the as-prepared Ni(OH)₂(B) sample. Scale bar, 50 nm. (c) XRD pattern of the as-prepared Ni(OH)₂(A) sample. (d) XRD pattern of the as-prepared Ni(OH)₂(B) sample. (e) STEM image of the as-synthesized Pt/Ni(OH)₂(A) sample. Scale bar, 100 nm. (f) STEM image of the as-synthesized Pt/Ni(OH)₂(B) sample.

Comment 3. Generally speaking the structure determination (or the discussion given) of $\text{Ni}(\text{OH})_x$ with unique cationic defects is not fully convincing. For example what can one say from the SAED pattern Figure S1? I do not know how the authors conclude the poly-“crystalline” structure from the results. Actually, XANES and EXAFS at Ni should be used to clarify this point further, in my opinion. Also the XRD patterns (Fig. 3b) should be well discussed in comparison to that of perfect- $\text{Ni}(\text{OH})_x$ case (Fig. 7b) where one could extract the information about preferential orientation of the NB. Also, Raman could add further information about the materials.

Response: We appreciate you for this insightful and constructive recommendation on the structure determination of our $\text{Ni}(\text{OH})_x$ NB material. To further clarify the polycrystalline and defect-rich structure of $\text{Ni}(\text{OH})_x$ NBs, we fitted the Fourier transform EXAFS spectra at the Ni K-edge of the perfect $\text{Ni}(\text{OH})_2$ and the $\text{Ni}(\text{OH})_x$ NBs to get more information on their different atomic structure. The fitting analysis results are shown in Fig. 4a in the revised manuscript and Supplementary Table 2 in the revised supplementary information (see below).

Figure 4 | Investigation of cation vacancies on $\text{Ni}(\text{OH})_x$ NBs. (a) Ni K-edge Fourier transform EXAFS spectra and corresponding fitting analysis for $\text{Ni}(\text{OH})_x$ NBs and the perfect $\text{Ni}(\text{OH})_2$.

Supplementary Table 2 | Structural parameters obtained from the EXAFS fitting analysis using $\beta\text{-Ni}(\text{OH})_2$ as the reference compound ($S_0^2 = 0.85$).

Sample	Path	N	R (Å)	σ^2 (10^{-3}Å^2)	ΔE_0 (eV)
$\text{Ni}(\text{OH})_x$ NBs	Ni-O	5.9 ± 0.5	2.06 ± 0.01	8.2 ± 1	-4.0 ± 0.1
	Ni-Ni	4.8 ± 0.5	3.12 ± 0.01	17 ± 2	-5.5 ± 0.3
perfect $\text{Ni}(\text{OH})_2$	Ni-O	6.1 ± 0.5	2.06 ± 0.01	6.0 ± 1	-3.8 ± 0.1
	Ni-Ni	6.2 ± 0.5	3.12 ± 0.01	10.4 ± 2	-2.4 ± 0.1

It can be seen that the coordination number (N) of the first Ni-Ni shell of Ni(OH)_x NBs is significantly lower than that of the perfect Ni(OH)₂, whereas their coordination numbers of the first Ni-O shell are nearly same, which indicates the formation of Ni²⁺ vacancies in Ni(OH)_x NBs. Besides, the values of Debye-Waller factor (σ^2) for the first Ni-O and Ni-Ni shells of Ni(OH)_x NBs are both higher than that of the perfect Ni(OH)₂, suggesting a higher degree of disorder in Ni(OH)_x NBs, which is in accord with the polycrystalline structure of Ni(OH)_x NBs. These results have also been described in the revised manuscript as follows:

p. 11 & 12 “As shown in Fig. 4a, the Ni K-edge Fourier transformed EXAFS spectrum of Ni(OH)_x NBs exhibits an apparent difference in spectral shape compared with that of the perfect Ni(OH)₂, implying the different local atomic arrangement and a defective structure of Ni(OH)_x NBs⁴⁸. Further EXAFS fitting analysis revealed that the values of Debye-Waller factor (σ^2) for the first Ni-O and Ni-Ni shells of Ni(OH)_x NBs are both higher than that of the perfect Ni(OH)₂, suggesting a higher degree of disorder in Ni(OH)_x NBs, which is in accord with the polycrystalline structure of Ni(OH)_x NBs (Supplementary Table 2). More importantly, the coordination number (N) of the first Ni-Ni shell of Ni(OH)_x NBs is about 4.8, which is significantly lower than that of the perfect Ni(OH)₂ (~ 6.2), whereas their coordination numbers of the first Ni-O shell are nearly same (~ 6.0), indicating the formation of Ni²⁺ vacancies in Ni(OH)_x NBs”

To gain more insight into the structure of the Ni(OH)_x NB sample, as you suggested, we further discussed its XRD pattern in comparison to that of the perfect Ni(OH)₂, which have been presented in Supplementary Fig. 2 in the revised supplementary information as follows:

Supplementary Figure 2 | XRD patterns of Ni(OH)_x NBs in comparison to the perfect Ni(OH)₂. It can be seen that all the diffraction peaks of Ni(OH)_x NBs are similar to that of the perfect Ni(OH)₂, which are consistent with a hexagonal layered structure α -Ni(OH)₂•0.75H₂O with lattice parameters of $a = b = 3.08 \text{ \AA}$ and $c = 23.41 \text{ \AA}$ (JCPDS 38-0715)¹. The strong diffraction peak from (012) planes clearly reveals that Ni(OH)_x NBs possess a preferred (012) orientation.

As for the Raman measurements, we performed at room temperature on a laser micro-Raman spectrometer (Horiba JY-HR-800) employing an argon-ion laser with an incident wavelength of 514 nm. The Raman spectrum of Ni(OH)_x NBs and the perfect are shown in Fig. R3, and it shows that a clear Raman spectrum of the perfect Ni(OH)₂ is obtained, but no obvious Raman signals can be detected on Ni(OH)_x NBs. Therefore, we failed to obtain more information on the structure of

Ni(OH)_x NBs from the Raman measurements.

Figure R3. Raman spectra of the perfect Ni(OH)₂ and Ni(OH)_x NB samples.

Comment 4. (EXAFS) The FT of the EXAFS is shown, but the quality of the analysis is not clear from the data shown. The spectra in *k*-space should be presented in ESI to evaluate the quality.

Response: Thanks for your suggestion. The EXAFS spectra in *k*-space have been presented in Supplementary Fig. 5 in the revised supplementary information as follows:

Supplementary Figure 5 | Corresponding EXAFS in *k*-space at the Pt-L₃ edge. The *k*³-weighted EXAFS in *k*-space for the Pt foil (a), PtO₂ (b) and Pt₁/Ni(OH)_x (c) at the Pt-L₃ edge.

Comment 5. Catalytic test using only the Ni(OH)_x NB support should be evaluated. Also, catalytic results using Pt supported on “perfect” Ni(OH)_x under the same condition as well as only support are missing.

Response: Thank you for this suggestion. As you suggested, we have evaluated the catalytic performance of the Ni(OH)_x NBs, perfect Ni(OH)₂, and Pt/Ni(OH)₂ under the same reaction condition of Pt₁/Ni(OH)_x. The results are shown in Supplementary Table 1 in the revised supplementary information as follows:

Supplementary Table 1 | Control experiments for diboration reactions.

Entry	Cat.	Conv. (%) ^a	Sel. (%) ^b	TOF (h ⁻¹)
1	Pt ₁ /Ni(OH) _x	97	99	3233
2	Ni(OH) _x NBs	0	0	0
3	Pt/Ni(OH) ₂	22	99	680
4	perfect Ni(OH) ₂	0	0	0

Standard reaction conditions: substrate **1a** (0.50 mmol) and **2a** (0.50 mmol), catalyst: Ni(OH)_x NB (6.5 mg), Pt₁/Ni(OH)_x (6.5 mg, Pt/substrate = 0.1%), Ni(OH)₂ (10.8 mg), Pt/Ni(OH)₂ (10.8 mg, Pt/substrate = 0.1%), mesitylene (2.0 mL) as solvent, $T = 120$ °C, $t = 0.3$ h. ^a Determined by gas chromatography (GC) analysis with dodecane as internal standard. ^b Determined by GC-MS analysis.

It is apparent that the diboration reaction fails at the absence of Pt, and the Pt/Ni(OH)₂ suffers from a relatively low reaction efficiency because the Pt species on the perfect Ni(OH)₂ mainly exist as nanoparticles. To describe these results, we have added the following sentences to the revised manuscript:

p. 8 “In contrast, no reaction occurred when using Ni(OH)_x NBs as the catalyst without Pt (Supplementary Table 1, entry 2).”

p. 11 “which resulting in a relatively low catalytic efficiency of Pt/Ni(OH)₂ for diboration reactions (Supplementary Table 1, entry 3 and 4).”

Comment 6. IR spectrum of “perfect” Ni(OH)_x?

Response: Thank you for this comment. The IR spectrum of the perfect Ni(OH)₂ is shown in Supplementary Fig. 10c as follows:

Supplementary Figure 10 | Characterization of the as-synthesized perfect Ni(OH)₂ material.
(c) The FT-IR spectrum of the perfect Ni(OH)₂.

Comment 7. What was the motivation to use the specific Pt precursor?

Response: Thanks for your question. There are **three reasons** for choosing H₂PtCl₆ as the Pt precursor in the preparation of the Pt₁/Ni(OH)_x catalyst: **(1)** The H₂PtCl₆ dissolves easily in alcohol, which is beneficial to the synthesis of the Pt₁/Ni(OH)_x catalyst with different Pt loading contents by the wet impregnation method; **(2)** When dissolving in alcohol, the H₂PtCl₆ can ionize to [PtCl₆]²⁻ anions, which are easily absorbed to the Ni(OH)_x support in the impregnation process; **(3)** The “Cl” coordination anion of H₂PtCl₆ can be easily removed from the Pt atom under the reduction with hydrogen at low temperature (100 °C) to provide the final Pt₁/Ni(OH)_x catalyst.

Comment 8. XPS spectrum of Ni(OH)_x NB is shown in Fig. 3b and that of “perfect”-Ni(OH)_x in Fig. S9. The deconvoluted peaks look quite different, whereas the spectra look quite similar. It is speculated that different deconvolution procedures may have been used, causing the difference. It would be necessary to show the two spectra by overlaying them to be precise about the interpretation.

Response: Thanks for your suggestion. The XPS spectra of Ni 2p_{3/2} peaks of Ni(OH)_x NBs and the perfect Ni(OH)₂ are overlaid in Fig. R4, in which a distinct difference between the XPS spectra can be observed.

Figure R4. XPS spectra of Ni 2p_{3/2} peaks of Ni(OH)_x NBs and the perfect Ni(OH)₂.

In order to make the XPS analysis more precise, we repeated the deconvolution procedures with same fitting parameters, and the results are shown in Fig. 4b in the revised manuscript (see below). Correspondingly, we have modified the manuscript to describe these results as follows:

p. 12 “Hence, we carried out X-ray photoelectron spectrum (XPS) measurements to detect the Ni³⁺ ions in the Ni(OH)_x NBs. Figure 4b displays the representative XPS spectrum in Ni 2p_{3/2} region of Ni(OH)_x NBs and the perfect Ni(OH)₂, which can be deconvoluted into four peaks. The signal of Ni³⁺ ions can be clearly distinguished from that of Ni²⁺ ions (centered at 855.3 and 861.0 eV) with higher binding energies at 857.2 and 864.7 eV, which correspond with the data reported⁵⁰⁻⁵². Distinctly, unlike the perfect Ni(OH)₂, Ni(OH)_x NBs display a stronger signal of Ni³⁺ ions, manifesting the possession of more Ni³⁺ ions in the Ni(OH)_x NBs.”

Figure 4 | Investigation of cation vacancies on Ni(OH)_x NBs. (b) XPS analysis of Ni(OH)_x NBs and the perfect Ni(OH)₂ in Ni 2p_{3/2} region.

Comment 9. (Fig. 3c & page 11. Line 188) XPS signal should neither be “absorption” nor “adsorption”.

Response: Thank you for this comment. Actually, Figure 3c in the manuscript represents the **soft X-ray absorption spectroscopy (sXAS) spectra** of samples, which **differs from the XPS spectra**. To make the description more accurate, we have corrected the Y axis title “Absorption (a. u.)” into “**Intensity (a.u.)**” and the “increased adsorption intensity” in the Line 188 of page 11 in the manuscript into “**increased intensity**” in the revised manuscript (see below).

Figure 4 | Investigation of cation vacancies on Ni(OH)_x NBs. (c) O K-edge sXAS spectra of Ni(OH)_x NBs and the perfect Ni(OH)₂.

p. 12 “As shown in Fig. 4c, the obviously **increased intensity** at 532.0 eV (Ni-O interaction) at O K-edge of Ni(OH)_x NBs relative to that of the perfect Ni(OH)₂ suggests that electrons transfer intensively from oxygen to nickel”

Comment 10. (Fig. 3c) The enhanced intensity of 532 eV is noted, but Fig. S2 is discussing about 531.0 eV assigned to oxygen atoms with slightly different origin (or expressed in different nuances). Clarification is needed.

Response: Thanks for your suggestion. We fear our plot may have misled you. The spectra shown in Fig. 3c (correspond to Fig. 4c in the revised manuscript) **belong to soft X-ray absorption spectroscopy (sXAS)**, and the abscissa represents the **photon energy, differing from the binding energy in XPS spectra**. Therefore, the enhanced intensity of 532 eV noted in the O K-edge sXAS spectra in Fig. 3c is irrelevant to the O 1s peak at 531.0 eV in the XPS spectrum in Fig. S2 (correspond to Supplementary Fig. 3 in the revised supplementary information), although the values are very close.

Comment 11. (page 3, line 54) “crabon” should be “carbon” I guess.

Response: We sincerely thank you for careful reading. We have carefully checked the manuscript and corrected the errors accordingly (see below).

p. 3 “the diboration of **carbon-carbon** multiple bonds represents a straightforward and atom-economic strategy”

Reviewer #2:

The authors prepared cation-rich Ni(OH)_x nanobelts and used them to support single-atom Pt species for catalytic diboration of alkynes and alkenes. They found very high catalytic activity for the reaction and attributed it to the strong interaction of Pt at the cation vacancies of the support. They further characterized their catalysts with TEM, XPS, EXAFS, UV-vis, etc. They also performed DFT calculations to show the strong interaction between Pt and the cation-rich Ni(OH)_x nanobelts. I have the following comments for the authors to address before I can recommend its publication.

Comment 1. *The authors' claim that their catalyst's turnover frequency is "60-fold higher than other reported heterogeneous catalysts" is very misleading because the reactions tested in "other reported heterogeneous catalysts" are very different from the present diboration reaction. For an apple-to-apple comparison, the authors should test the same reaction, e.g. CO oxidation, CO₂ hydrogenation, or other hydrogenation reactions tested before.*

Response: Thank you very much for this comment. In fact, the "**main pursuit**" of this manuscript is to take advantage of isolated single-atomic-site Pt catalysts to improve the catalytic efficiency of diboration reactions. Our ultimate target is to develop new isolated single-atomic-site catalysts for opening up more opportunities in **heterogenization of homogeneous catalytic reactions**. From this point of view, therefore, we consider it more meaningful to choose the diboration reaction as the model reaction for our Pt₁/Ni(OH)_x catalyst.

More importantly, this Pt₁/Ni(OH)_x catalyst exhibits its **catalytic specificity for diboration reactions with the help of the Ni²⁺ vacancies** on the Ni(OH)_x support, which was disclosed by further DFT calculations on the catalytic mechanism of Pt₁/Ni(OH)_x for diboration reactions (for details, see Supplementary Computational Methods section in the revised supplementary information). Besides, in our previous work, the Pt₁/Ni(OH)_x catalyst has already been tested in other reactions, but its catalytic performance for these reactions is not good enough. For example, the result of **CO oxidation** catalyzed by Pt₁/Ni(OH)_x is shown in Fig. R5 as below. The Pt₁/Ni(OH)_x catalyst displays its best activity at 100 °C with a corresponding conversion of CO at less than 30%, which represents a very general catalytic performance.

Figure R5. Catalytic performance evaluation of Pt₁/Ni(OH)_x for CO oxidation.

In this work, the TOF value of Pt₁/Ni(OH)_x was calculated and compared on a **same model reaction** (diboration of phenylacetylene) with other reported heterogeneous catalysts. However, for an apple-to-apple comparison, it is indeed required the same reaction condition. **As for liquid organic reactions, different catalysts are usually suitable in different reaction conditions.** Taking the reaction of diboration of phenylacetylene as an example, the reported heterogeneous catalysts adapted at different temperature (70 – 130 °C) with different solvent (neat or toluene). In this work, to make comparisons as accurate as possible, we have choose the **representative conditions** of diboration reactions reported in the literature as our standard conditions for Pt₁/Ni(OH)_x. Nonetheless, to make the description of the activity of our Pt₁/Ni(OH)_x catalyst more precise, we have modified the manuscript as follows:

p. 2 “For the diboration of phenylacetylene, the catalyst yields an overall turnover frequency upon reaction completion of ~3,000 h⁻¹, **much** higher than other reported heterogeneous catalysts.”

p. 4 “A TOF_{overall} upon reaction completion **much greater than that of all reported heterogeneous catalysts** is demonstrated on Pt₁/Ni(OH)_x in the diboration of alkynes.”

p. 8 “The calculated TOF_{overall} value of Pt₁/Ni(OH)_x upon this reaction completion can reach a high level as ~ 3,000 hours⁻¹, **much** higher than that of other heterogeneous catalysts reported previously.”

Comment 2. *Can the authors show the results for the control, i.e., the cation-rich Ni(OH)_x nanobelts as a catalyst without Pt?*

Response: Thanks for your suggestion. The results of control experiments have been shown in Supplementary Table 1 in the revised supplementary information (see below). Apparently, the diboration reaction fails at the absence of Pt, and the isolated single-atomic-site Pt exhibits a much higher catalytic activity than Pt nanoparticles.

Supplementary Table 1 | Control experiments for diboration reactions.

Entry	Cat.	Conv. (%) ^a	Sel. (%) ^b	TOF (h ⁻¹)
1	Pt ₁ /Ni(OH) _x	97	99	3233
2	Ni(OH) _x NBs	0	0	0
3	Pt/Ni(OH) ₂	22	99	680
4	perfect Ni(OH) ₂	0	0	0

Standard reaction conditions: substrate **1a** (0.50 mmol) and **2a** (0.50 mmol), catalyst: Ni(OH)_x NB (6.5 mg), Pt₁/Ni(OH)_x (6.5 mg, Pt/substrate = 0.1%), Ni(OH)₂ (10.8 mg), Pt/Ni(OH)₂ (10.8 mg, Pt/substrate = 0.1%), mesitylene (2.0 mL) as solvent, *T* = 120 °C, *t* = 0.3 h. ^a Determined by gas chromatography (GC) analysis with dodecane as internal standard. ^b Determined by GC-MS analysis.

Comment 3. The DFT results of stronger binding of Pt on the cation-rich Ni(OH)_x nanobelts are expected, so they add limited value to the present work. It'd be more valuable if the authors could use DFT to further reveal the catalytic mechanism.

Response: We appreciate you for this valuable recommendation. Following your recommendation, we have investigated the catalytic mechanism of the Pt₁/Ni(OH)_x catalyst for diboration reactions by performing DFT calculations, and a detailed description of the results is presented in the Supplementary Computational Methods in the revised supplementary information as follows:

The catalytic mechanism of diboration reactions happening on single Pt atom has been proposed before¹⁵⁻¹⁷. As shown in Supplementary Fig. 12, a general cycle of Pt-catalyzed diboration reaction contains three steps as (1) oxidative addition of B-B bond to Pt, (2) insertion of C-C multiple bond and (3) reductive elimination of C-B bond from Pt.

Supplementary Figure 12 | General cycle of Pt-catalyzed diboration reactions.

The diboration of phenylacetylene with bis(pinacolato)diboron (B₂pin₂) was used as the model reaction. According to the above reaction pathway, we calculated the energy evolution and reaction barrier of three crucial steps (Supplementary Fig. 13). The first step of diboration reaction contains the adsorption of B₂pin₂ and the dissociation of B-B bond, with an energy decrease of 3.49 eV. The energy barrier of dissociation of B-B bond is about 0.29 eV, which is easy to overcome in experiment. Therefore, the addition of B-B bond to isolated Pt atom is energetic and kinetic favorable in experiment. It is worth noting that not only the Ni²⁺ vacancies play an important role in locating isolated Pt atoms, but also the low-coordination oxygen atoms around the located Pt atoms benefit the dissociation of B-B bond. In the second step, the reaction process contains the adsorption of phenylacetylene and the insertion of Bpin on acetylenic bond, with an energy decrease of 1.41 eV. The barrier of insertion of Bpin on acetylenic bond is 1.57 eV, which is crucial process in the second step. The third step contains the insertion of another Bpin on ethylenic bond and the desorption of reacted chemical group, with an energy increase of 2.54 eV, indicating that it is an endothermic process. The barrier of insertion of another Bpin is 1.40 eV, which is consistent with that of the insertion of Bpin in the second step. To summarize, the overall diboration reaction of phenylacetylene with B₂pin₂ on Pt₁/Ni(OH)_x is an exothermic reaction with an energy decrease of 2.36 eV and the insertion of the Bpin on acetylenic bond is the rate-limiting step for diboration reaction.

Supplementary Figure 13 | Energy evolution and reaction barrier of three proposed crucial steps of $\text{Pt}_1/\text{Ni}(\text{OH})_x$ catalyzed diboration reactions.

We also added the following sentence in the revised manuscript to mention the important conclusion of the above DFT calculations:

p. 14 “Further DFT calculations on the catalytic mechanism of the $\text{Pt}_1/\text{Ni}(\text{OH})_x$ catalyst for diboration reactions even disclosed that the Ni^{2+} vacancies not only play an important role in locating isolated Pt atoms, but also are conducive to the diboration reactions because the low-coordination oxygen atoms at the vacancy site around the located Pt atoms benefit the dissociation of B-B bond (for details, see Supplementary Computational Methods section).”

Reviewer #3:

This article describes a synthesis of diborylated alkenes or alkanes using heterogeneous catalysis. It is interesting. It needs some proof reading to avoid errors such as "crabon carbon" and "dibration."

Response: Thank you very much for careful reading. We feel sorry for our carelessness, we have carefully checked our manuscript. The spelling errors and typos has been corrected in the revised manuscript as follows:

- p. 3 “crabon-carbon multiple bonds” has been replaced by “carbon-carbon multiple bonds”
- p. 4 “nonporous-gold” has been replaced by “nanoporous-gold”
- p. 8 “dibration reactions” has been replaced by “diboration reactions”
- p. 8 “hydroborated product” has been replaced by “hydroborylated product”
- p. 12 “increased adsorption intensity” has been replaced by “increased intensity”
- p. 15 “in literature” has been replaced by “in the literatures”
- p. 16 “in literature” has been replaced by “in the literature”

For publication, apart from the English revision, I would expect to see the following:

Comment 1. *Table 1: Proper characterisation of the diborylated products. Yields quoted are by GC and as conversions. I would like to see both NMR characterisation (^1H , ^{13}C , with scanned spectra in the ESI) and isolated (after chromatography) yields on > 1 mmol scale reactions.*

Response: Thank you very much for this comment. The NMR characterization (^1H & ^{13}C NMR) of the diborylated products and corresponding scanned spectra have been supplemented to the **Characterization of Products** and **NMR Spectra of Products** sections in the revised supplementary information. In addition, the experiment of expanding the scale of the diboration reaction over $\text{Pt}_1/\text{Ni}(\text{OH})_x$ has been conducted, and the result is present in Supplementary Fig. 9 in the revised supplementary information (see below). It shows that although the reaction scale increases to 2.0 mmol, the corresponding product can also be obtained in 89% isolated yield.

Supplementary Figure 9 | Expansion of the scale of the diboration reaction over $\text{Pt}_1/\text{Ni}(\text{OH})_x$. Standard reaction conditions: substrate 1 (2.0 mmol) and 2 (2.0 mmol), catalyst: $\text{Pt}_1/\text{Ni}(\text{OH})_x$ (26.5 mg, Pt/substrate = 0.1%), mesitylene (8.0 mL) as solvent, $T = 120$ °C, $t = 0.3$ h. The product 3 was isolated in 93 % yield by flash column chromatography on silica gel with *n*-hexane/ethyl acetate as eluent.

Accordingly in the revised manuscript, we added the following sentence to describe the above result:

- p. 9 “Moreover, the expansion of the reaction scale has no effect on the catalytic efficiency of this

Pt₁/Ni(OH)_x catalyst for such diboration reactions (Supplementary Fig. 9).”

I'd say accept with minor corrections.

Comment 2. *There are only 5 examples in Table 1, which is hardly demonstrating reaction scope, substrate, functional group tolerance. I'd expect to see >10 examples including other boronate esters, other alkynes including more alkyl, heterocyclic, sterically and electronically diverse aryl derivatives to show yields vs steric and electronic effects.*

Response: We sincerely appreciate your valuable comment. As can be seen from the Fig.3 in the revised manuscript, we have expanded the substrate scope of Pt₁/Ni(OH)_x catalyzed diboration reactions, and **15 examples** including different borate esters and alkynes are taken to investigate the steric and electronic effects of the reactions, which have been well discussed in the revised manuscript (see below). In addition to all the alkynes above, some heterocyclic alkynes such as 2-pyridylacetylene and 2-thienylacetylene have also been tested with Pt₁/Ni(OH)_x in the diboration reactions. However, only trace conversions of the substrates and almost no products were detected even though extending the reaction time to 6 h.

p. 9 & 10 “We further investigated the substrate scope of the diboration reactions to study the influence of substrate categories on the catalytic efficiency of Pt₁/Ni(OH)_x. As shown in Fig. 3, the aryl alkynes bearing electron-donating groups (R = Me, OMe) can react smoothly with B₂pin₂ over Pt₁/Ni(OH)_x at the same conversion rate of phenylacetylene, affording the corresponding products **3ba** and **3ca** in excellent yields. When the substituents are electron-withdrawing groups (R = Cl, Br, NO₂) on the aryl ring, however, a longer reaction time is required to give the target molecular **3da-3fa** in high yields. Differently from substituent types, substituent positions have no influence on the catalytic efficiency of Pt₁/Ni(OH)_x, whether ortho-methyl or meta-methyl substituted phenylacetylene can completely transform into to the products **3ga** and **3ha** without any loss in the reaction rate. As for diboration of internal aryl alkynes like diphenylacetylene, Pt₁/Ni(OH)_x suffers from a relatively low catalytic activity, although a complete conversion of the substrate and a quantitative selectivity of diborylated product **3ia** can be also achieved. To our delight, different kinds of aliphatic alkynes are appropriate for the diboration reactions over Pt₁/Ni(OH)_x as well, furnishing the desired products **3ja-3la** with the similar reaction efficiency to the terminal aryl alkynes. Besides the various alkynes, different boronate esters like bis(neopentylglycolate)diboron (B₂neop₂), can also work well with phenylacetylene to provide the product **3ab** in excellent yield and selectivity at the same reaction rate of B₂pin₂. Even more, alkenes were also chosen as substrates to evaluate the catalytic performance of Pt₁/Ni(OH)_x for diboration reactions. It shows that the diboration of styrene and 1-octene catalyzed by Pt₁/Ni(OH)_x can proceed well and provide a selectivity to products **3ma** and **3na** of 99% at the conversion level of 90% and 86%, respectively, although the reaction rates are lower than that of alkynes with similar molecular structures.”

Figure 3 | Substrate scope of diboration reactions over the $\text{Pt}_1/\text{Ni}(\text{OH})_x$ catalyst. Standard reaction conditions: substrate 1 (0.50 mmol) and 2 (0.50 mmol), $\text{Pt}_1/\text{Ni}(\text{OH})_x$ catalyst, Pt/substrate = 0.1%, mesitylene (2.0 mL) as solvent, $T = 120\text{ }^\circ\text{C}$, $t = 0.3\text{ h}$. Conversion are determined by gas chromatography (GC) analysis with dodecane as internal standard. Selectivities are determined by GC-MS analysis. ^a $t = 1.0\text{ h}$. ^b $t = 6.0\text{ h}$ and substrate 2 (0.75 mmol) was used.

Comment 3. Many homogeneous reference citations are missing-it would be nice to cite some recent reviews e.g. Coord. Chem. Rev. 2017, 336, 54; Chem. Rev. 2016, 116, 9091–9161.

Response: Thanks for your suggestion. As you suggested, correlative references about homogeneous diboration reactions have been added into the revised manuscript as follows:

Ref. 31: Ansell, M. B., Navarro, O. & Spencer, J. Transition metal catalyzed element–element' additions to alkynes. *Coord. Chem. Rev.* **336**, 54-77 (2017).

Ref. 32: Neeve, E. C., Geier, S. J., Mkhaliid, I. A., Westcott, S. A. & Marder, T. B. Diboron(4) Compounds: From Structural Curiosity to Synthetic Workhorse. *Chem. Rev.* **116**, 9091-9161 (2016).

Reviewers' comments:

Reviewer #1 (Remarks to the Author):

The authors revised the manuscript with a great care and improved the quality significantly. All raised questions have been satisfactorily answered and I recommend this manuscript for publication in Nature Communications.

Reviewer #2 (Remarks to the Author):

the authors have addressed my previous comments. i now recommend publication of this paper as is.

Reviewer #3 (Remarks to the Author):

This is a greatly enhanced submission and should be ready for publication with just a few very minor corrections (without any more review from me).

I'm very happy that the authors have put in a great deal of effort to show the impressive scope of this reaction as it does give it more value to the scientific community especially in terms of diboronated alkenes that can be made.

I would just urge the following:

Refs 14-18 on diboration should also be in the main text, not just ESI.
Isolated yields should be given for products (after purification, not just GC conversion), a number of typos need addressing in the ESI e.g. picks (peaks) and a few more.

Then, I feel it is a Nat Comm!

Responses to Reviewers:

The reviewers' comments are laid out below in *Italic font* and specific concerns have been numbered. Our responses are given in normal font and changes/additions to the manuscript and supplementary information are highlighted by using **blue colored text**.

Following is our point to point answer to all the comments:

Reviewer #1:

The authors revised the manuscript with a great care and improved the quality significantly. All raised questions have been satisfactorily answered and I recommend this manuscript for publication in Nature Communications.

Response: We appreciate your recommendation of acceptance and helpful comments in the reviewing process and are pleased to have our manuscript be reviewed by you.

Reviewer #2:

the authors have addressed my previous comments. i now recommend publication of this paper as is.

Response: Thank you very much for your valuable comments in the reviewing process, which have helped us to improve the quality of the whole manuscript. We sincerely appreciate you for recommending our manuscript be accepted by *Nature Communications*.

Reviewer #3:

This is a greatly enhanced submission and should be ready for publication with just a few very minor corrections (without any more review from me).

I'm very happy that the authors have put in a great deal of effort to show the impressive scope of this reaction as it does give it more value to the scientific community especially in terms of diboronated alkynes that can be made.

I would just urge the following:

Response: We sincerely appreciate your positive review and recommendation that our manuscript be accepted! The answers to the remaining issues you raised are detailed here.

Comment 1. Refs 14-18 on diboration should also be in the main text, not just ESI.

Response: Thanks for your suggestion. The references on diboration in the ESI are actually Refs 15-18, and as you suggested, we have added them into the final revised manuscript as follows:

Ref. 29: Takaya, J. & Iwasawa, N. Catalytic, direct synthesis of bis(boronate) compounds. *ACS Catal.* **2**, 1993-2006 (2012).

Ref. 31: Ishiyama, T. *et al.* Platinum(0)-Catalyzed Diboration of Alkynes with Tetrakis(alkoxy)diborons: An Efficient and Convenient Approach to cis-Bis(boryl)alkenes. *Organometallics* **15**, 713-720 (1996).

Ref. 32: Ansell, M. B. *et al.* An experimental and theoretical study into the facile, homogeneous (N-heterocyclic carbene)₂-Pd(0) catalyzed diboration of internal and terminal alkynes. *Catal. Sci. Technol.* **6**, 7461-7467 (2016).

Ref. 33: Morgan, J. B. & Morken, J. P. Catalytic enantioselective hydrogenation of vinyl bis(boronates). *J. Am. Chem. Soc.* **126**, 15338-15339 (2004).

Comment 2. Isolated yields should be given for products (after purification, not just GC conversion)

Response: Thank you for this suggestion. In our previous work, we have tried to obtain the isolated yields of diborylated products in Fig. 3. However, it was found that these products could decompose slowly when isolated by flash column chromatography either on silica gel or basic alumina. In addition, some products, including **3ja**, **3ka**, **3la**, **3ma** and **3na**, were even hard to be monitored by thin layer chromatography (TLC). Therefore, the accurate isolated yields of the whole products were difficult to obtain and the corresponding GC conversions were given only. Based on our GC-MS and GC analysis results and the isolated yield obtained from the large scaled diboration reaction (Supplementary Fig. 9), we believed that the GC conversions were basically consistent with the real yields of the products.

Comment 3. a number of typos need addressing in the ESI e.g. picks (peaks) and a few more. Then, I feel it is a Nat Comm!

Response: Thank you very much for careful reading. The spelling errors and typos has been corrected in the final revised ESI as follows:

p. 3 “a board band” has been replaced by “a **broad** band”

p. 3 “confirmed” has been replaced by “**confirm**”

p. 5 “a main pick” has been replaced by “a main **peak**”

p. 5 “recoverd” has been replaced by “**recovered**”